# Isoleucilactucin Ameliorates Coal Fly Ash-Induced Inflammation through the NF-κB and MAPK Pathways in MH-S Cells

**DOI:** 10.3390/ijms22179506

**Published:** 2021-09-01

**Authors:** H. M. Arif Ullah, Tae-Hyung Kwon, SeonJu Park, Sung Dae Kim, Man Hee Rhee

**Affiliations:** 1Department of Veterinary Medicine, College of Veterinary Medicine, Kyungpook National University, Daegu 41566, Korea; arif55dml@knu.ac.kr (H.M.A.U.); kim79sd@knu.ac.kr (S.D.K.); 2Department of Research and Development, Chuncheon Bio-Industry Foundation (CBF), Chuncheon 24232, Korea; 3Chuncheon Center, Korea Basic Science Institute (KBSI), Chuncheon 24341, Korea; sjp19@kbsi.re.kr

**Keywords:** lung macrophages, isoleucilactucin, anti-inflammatory properties, NF-κB pathway, AMPK pathway

## Abstract

We investigated whether isoleucilactucin, an active constituent of *Ixeridium dentatum*, reduces inflammation caused by coal fly ash (CFA) in alveolar macrophages (MH-S). The anti-inflammatory effects of isoleucilactucin were assessed by measuring the concentration of nitric oxide (NO) and the expression of pro-inflammatory mediators in MH-S cells exposed to CFA-induced inflammation. We found that isoleucilactucin reduced CFA-induced NO generation dose-dependently in MH-S cells. Moreover, isoleucilactucin suppressed CFA-activated proinflammatory mediators, including cyclooxygenase-2 (COX2) and inducible NO synthase (iNOS), and the proinflammatory cytokines such as interleukin-(IL)-1β, IL-6, and tumor necrosis factor (TNF-α). The inhibiting properties of isoleucilactucin on the nuclear translocation of phosphorylated nuclear factor-kappa B (p-NF-κB) were observed. The effects of isoleucilactucin on the NF-κB and mitogen-activated protein kinase (MAPK) pathways were also measured in CFA-stimulated MH-S cells. These results indicate that isoleucilactucin suppressed CFA-stimulated inflammation in MH-S cells by inhibiting the NF-κB and MAPK pathways, which suggest it might exert anti-inflammatory properties in the lung.

## 1. Introduction

Inflammation is a critical aspect of the immune response against cell damage and pathogens, which also provides a defense mechanism to remove harmful stimuli [1,2,3]. Increased movement of plasma and leukocytes from the blood into wounded regions causes acute inflammation, which is an initial response of the organism to damaging stimuli. When it becomes chronic, inflammation causes a gradual shift in the cell types present at the site of inflammation and can result in a destruction or regeneration of tissues, depending on the context [4]. Dysregulated inflammation is associated with various chronic diseases [2,5]. Macrophages play a crucial role in inflammatory processes, primarily by producing proinflammatory mediators that include nitric oxide (NO), inducible NO synthase (iNOS), cyclooxygenase-2 (COX-2), as well as the cytokines interleukin (IL)-1β, IL-6, and tumor necrosis factor-α (TNF-α) [6,7]. Therefore, inhibiting proinflammatory macrophage transformation is considered a critical strategy in the treatment of numerous inflammatory disorders.

In the last few decades, environmental contaminants were shown to cause immunologic dysfunction in chronic respiratory inflammatory disorders: asthma, chronic respiratory diseases, chronic obstructive pulmonary disease (COPD), and various types of lung cancer [1,6,8,9,10]. COPD is the world’s third leading cause of death, with 3 million individuals dying from the disease in 2016 [11]. In modern industrial countries, particulate matter (PM) represents the most dangerous air contaminant [12]. The main sources of PM include outdoor pollutants from gasoline fuel burning in automobiles and industries, as well as indoor pollutants from cigarette smoke and culinary exhausts [6,13]. According to previous research, exposure to PM has been linked to worsening of asthma symptoms, as well as increased hospitalization and mortality in individuals with COPD [14,15]. Exposure to PM could induce alveolar macrophages to produce excessive amounts of proinflammatory mediators, such as iNOS, COX2, TNF-α, IL-1β, and IL-6, which determine a wide range of pathological and clinical symptoms [9,16].

The pharmaceutical industry’s interest in natural drugs has grown, thus including natural resources in drug development [3,17,18]; indeed, over 100 medications derived from natural products are now being tested in clinical trials [3,19]. Isoleucilactucin, one of the major active constituents of *Ixeridium dentatum*, has been reported to have amylase secretion activity after treatment with high glucose in human salivary gland cells [20]. Coal fly ash (CFA) is made up of a variety of liquid and solid PM, including asbestos, coal, and combustion particles of various sizes and origins [1,6,21]. However, the effect of isoleucilactucin in the CFA-stimulated inflammatory cascade on alveolar macrophages (MH-S) remains elusive. The goal of this study is to determine whether isoleucilactucin has anti-inflammatory activities upon CFA-induced inflammatory stimulation of MH-S cells.

## 2. Results

### 2.1. Isoleucilactucin Protected against CFA-Induced Nitric Oxide Production and Cell Death in Alveolar Macrophage (MH-S) Cells

Nitric oxide (NO) is a vital regulator in the inflammatory process, while excess NO contributes to the development of numerous inflammatory disorders [22,23,24]. In our study, the NO levels in murine MH-S cells in response to CFA stimulation were assessed using the Griess reaction. Isoleucilactucin decreased NO induction potently and dose-dependently (Figure 1A).

The 3-(4,5-dimethylthiazol-2-yl)-2,5-diphenyltetrazolium bromide (MTT) test was used to demonstrate cell viability, and the data showed that isoleucilactucin had no effect on cell toxicity at the different concentrations examined in comparison to the control group (Figure 1B). Our findings together indicate that isoleucilactucin suppressed NO generation dose-dependently and at noncytotoxic levels.

### 2.2. Suppressive Effect of Isoleucilactucin on CFA-Induced Proinflammatory Cytokines in MH-S Cells

To evaluate the anti-inflammatory effect of isoleucilactucin on CFA-activated inflammation in the MH-S cells, reverse transcription polymerase chain reaction (RT-PCR) was assessed. After 30 min of pretreatment with isoleucilactucin, the levels of CFA-induced proinflammatory factors were decreased in MH-S cells. mRNA levels of proinflammatory mediators, such as iNOS and COX-2, as well as proinflammatory cytokines, such as IL-1β, IL-6, and TNF-α, were measured by RT-PCR to explore the anti-inflammatory effects of isoleucilactucin (Figure 2A). At the mRNA level, proinflammatory factors were found to be dose-dependently decreased (Figure 2B–F). RT-PCR results showed that isoleucilactucin prevented CFA-induced inflammatory cytokine production and reduced the expression levels of proinflammatory factors.

### 2.3. Isoleucilactucin Ameliorated CFA-Induced mRNA Expression of Proinflammatory Cytokines in MH-S Cells

To validate the RT-PCR results in MH-S cells, proinflammatory mediators mRNA expression was next investigated. Following isoleucilactucin (12.5, 25, 50, and 100 μM) administration, the mRNA expression levels of the proinflammatory mediators were markedly and dose-dependently inhibited (Figure 3A–E). The real-time PCR results demonstrated that isoleucilactucin significantly lowered the proinflammatory factors in a concentration-dependent manner (Figure 3A–E).

### 2.4. Isoleucilactucin Inhibited the Translocation of Nuclear Factor-Kappa B in CFA-Treated MH-S Cells

An immunofluorescence (IF) analysis was used to examine the translocation of activated nuclear factor (NF)-κB from the cytoplasm to the nucleus, to determine whether the anti-inflammatory actions of isoleucilactucin could be mediated by the NF-κB signaling pathway in CFA-stimulated alveolar macrophages. In activated MH-S cells, treatment with CFA increased NF-κB translocation from the cytoplasm to the nucleus, but treatment with the maximum dose of isoleucilactucin (100 μM) inhibited phosphorylated (p)-NF-κB nuclear translocation (Figure 4). Here, we used Bay-11 as an NF-κB inhibitor. Our IF experiment reveals that the anti-inflammatory effects of isoleucilactucin may result from partly inhibiting phosphorylation of the NF-κB signaling cascade.

### 2.5. Isoleucilactucin Inhibits the Activation of Nuclear Factor-Kappa B and Mitogen-Activated Protein Kinase Signaling Pathways in CFA-Treated MH-S Cells

To explore the molecular pathways underlying the protective effects of isoleucilactucin in our in vitro CFA-induced inflammation model, we performed Western blot analysis. CFA activates the inflammatory pathway, and the NF-κB and MAPK signaling pathways are important in inflammatory processes [25,26,27]. Pretreatment with isoleucilactucin greatly decreased the phosphorylation (p) of inhibitor of kappa B (IκB) and NF-κB in MH-S cells, whereas treatment with CFA significantly enhanced the p-IκB and p-NF-κB in alveolar macrophages (Figure 5A–C). Furthermore, as compared to treatment with CFA alone, the MAPK pathways, including phosphorylated extracellular signal-regulated kinase (p-ERK), phosphorylated c-Jun N-terminal kinase (p-JNK), and p-P38, were dose-dependently reduced (Figure 5D–G). Collectively, these findings indicate that the CFA-induced activation of p-NF-κB, p-IκB, p-ERK, p-JNK, and p-P38 in MH-S cells was significantly reduced by pretreatment with isoleucilactucin.

## 3. Discussion

The inflammatory process involves an abnormally high production of NO caused by iNOS synthesis [28,29]. In the pathophysiology of inflammatory disorders, iNOS plays a crucial role in the release of NO [30,31]. Furthermore, inflammatory stimuli increase COX-2 during the inflammatory process [32,33]. In the current study, only the CFA-activated group produced more NO compared to the control group, whereas pretreatment with isoleucilactucin inhibited NO generation.

Throughout the inflammation process, endotoxins and cytokines cause rapid changes in NO gene expression, resulting in de novo recruitment of the iNOS and COX-2 pathways [34,35]. In the current study, the CFA-treated group displayed elevated mRNA levels of proinflammatory factors, iNOS, COX2, TNF-α, IL-1β, and IL-6. In addition, treatment with isoleucilactucin decreased the mRNA levels of the proinflammatory mediators, including cytokines. These findings indicate that isoleucilactucin might have anti-inflammatory properties. Our results support previous observations that the COX-2 and iNOS mRNA expression levels were elevated following CFA stimulation [1,6,8].

According to our IF staining data, CFA elevated p-NF-κB translocation into the nucleus, contrary to isoleucilactucin (100 μM) and Bay-11 (10 μM), which significantly reduced the translocation. In an earlier investigation, we discovered a similar outcome [6]. Bay-11 was also previously shown to have anti-inflammatory properties through inhibiting the phosphorylation of IκB [36,37,38].

The NF-κB and MAPK signaling cascades were identified as key players in the inflammatory process [39,40,41,42]. NF-κB, an important transcription factor, plays a critical role in the inflammatory response [43,44,45] and the MAPK signaling cascade is also important for the inflammatory process [27,46]. As a result, we investigated anti-inflammatory mechanisms mediated by isoleucilactucin in a macrophage cell line (MH-S). Western blotting was used to evaluate the levels of protein expression of p-IκB, p-NF-κB, p-ERK, T-ERK, p-JNK, T-JNK, p-P38, and T-P38. CFA treatment increased the activity of the NF-κB and MAPK signaling pathways in the MH-S cell line, but isoleucilactucin administration dramatically inhibited the NF-κB and MAPK signaling. Based on these results, we predicted that isoleucilactucin would reduce the IκB and NF-κB phosphorylation, resulting in a proinflammatory factor suppressive action. Our data show that isoleucilactucin reduced CFA-activated inflammation by lowering the NF-κB and MAPK signaling pathway activity levels (Figure 6).

## 4. Materials and Methods

### 4.1. Reagents

Roswell Park Memorial Institute medium (RPMI-1640) for culturing macrophage, penicillin–streptomycin, fetal bovine serum, and phosphate-buffered saline were acquired from Welgene (Gyeongsan-si, Korea). Oligo-dT and primer sequences, including COX-2, iNOS, IL-1β, IL-6, and TNF-α, were obtained from Bioneer (Daejeon, Republic of Korea). Other reagents were purchased from Invitrogen, Sigma-Aldrich, and Thermo Fisher.

### 4.2. Plant Material Collection

The leaves of *Ixeridium dentatum* collected in the area of Jeonju-si, Jeonbuk, Korea, in May 2017, and were certified by Professor Sang-Won Lee of the National Institute of Horticultural and Herbal Science, South Korea. A voucher specimen is kept in the Herbarium of the College of Pharmacy, Yonsei University, Incheon, Korea (ID201705).

### 4.3. Plan Material Extraction and Isolation

The plant extraction and isolation were conducted as previously described [20,47]. In brief, dried leaves of *I. dentatum* (5.0 kg) were extracted with MeOH (3 × 10 L, 50 °C) and sonicated for 4 h after solvent evaporation, producing 290.0 g extract. The extract was suspended in H_2_O and partitioned with n-hexane and EtOAc to obtain n-hexane (ID1, 91.0 g), EtOAc (ID2, 15.1 g), and H_2_O (ID3, 178.0 g) extracts after the solvents were removed in vacuo. In a Diaion HP-20 column, the H_2_O fraction was chromatographed with increasing MeOH concentrations (25, 50, and 75%) to acquire three sub-fractions: ID3A (14.5 g), ID3B (13.3 g), and ID3C (23.2 g). To obtain isoleucilactucin, the ID3C fraction was chromatographed on an RP-18 CC and eluted with MeOH:H_2_O (1:2, *v*/*v*) (5.2 mg). The resulting isoleucilactucin was a white amorphous powder. The HR-ESI-MS (Agilent 6530) [M–H_2_O]^+^ ion at m/z 389.1825 identified its molecular formula as C_21_H_29_NO_7_ (calculated for C_21_H_27_NO_6_, 389.1838).

### 4.4. Cell Culture and Treatment

According to a previously published method [6], murine alveolar macrophage cell line (MH-S) was grown in RPMI supplemented with heat-inactivated 10% FBS, 100-unit/mL penicillin, and 100-g/mL streptomycin (1% antibiotics). Cells were then cultured and incubated in a humidified incubator at 37 °C with 5% CO_2_.

### 4.5. Nitric Oxide Assay

In accordance with our earlier investigation [48], Griess reagent A and B were used to assess the NO concentration. MH-S were seeded and cultivated for 18 h in a 24-well plate with or without CFA (2.5 μg/mL) for the basal group (control), only CFA-treated group, and CFA with isoleucilactucin (12.5, 25, 50, and 100 μM) groups at the indicated concentrations. In total, 100 µL of the Griess reagents were mixed with 100 µL of cell culture supernatants and incubated at room temperature for 10 min. On a microplate reader (Versamax, Molecular Devices, San Jose, CA, USA), the absorbance was measured at 540 nm.

### 4.6. Cell Viability Assay

To investigate cytotoxicity, an experiment of cell viability was done as previously described [6,48], using a 10% MTT reagent at 900 µL/well in the culture media. The supernatants were removed after 3 h of incubation at 37 °C in 5% CO_2_. Wells were filled with 100% of DMSO (500 µL/well) and incubated for 10 min with shaking at room temperature. Finally, the absorbance was measured at 560 nm using a microplate reader (Versamax, Molecular Devices, CA, USA).

### 4.7. PCR Analysis

Using previously published methods, PCR analysis was performed [2,6,48]. The MH-S cells were nontreated or treated with isoleucilactucin (12.5, 25, 50, and 100 μM) for 30 min at the indicated doses and received CFA stimulation for 18 h. A TRIzol reagent was used to extract RNA from the cells. In total, 2 µg of total RNA were annealed for 10 min at 70 °C with oligo-dT, then cooled for 5 min on ice before reverse transcription (RT) in 20 µL of reaction mixture at 42.5 °C on a thermocycler for 90 min. To inactivate the reverse transcriptase, reaction was stopped at 95 °C for 5 min. cDNA from an RT reaction was used in a PCR premix to perform RT-PCR (Bioneer). On a 1% agarose gel stained with ethidium bromide (0.006%), the PCR products were also electrophoresed. To visualize the band, ImageQuant LAS 500 was used. The band density intensity was standardized using glyceraldehyde-3-phosphate dehydrogenase (GAPDH). SYBR green was used in the real-time PCR. Table 1 shows the primer sequences (Bioneer, Daejeon, Republic of Korea) for RT-PCR and real-time PCR.

### 4.8. Immunofluorescence Analysis

The immunofluorescence (IF) experiment was carried out as described previously [2,6]. Cells were washed in PBS before being fixed in paraformaldehyde (4 %) for 10 min. Cells were then permeabilized with triton X-100 (0.2 %) in TBS (TBST) for 10 min before being rinsed three times with TBST for 5 min each time. Using 2% BSA, cells were blocked for 1 h at room temperature before being incubated overnight at 4°C with the primary antibody rabbit anti-p-NF-κB (Cat. no: #3033). The cells were washed with TBST three times for 5 min each time. Samples were then incubated with secondary antibody, IgG Fab2 (Cat. no: #4413), for 1 h in the dark at room temperature, and washed three times with TBST before being mounted with a ProLong Gold Antifade Reagent with DAPI to visualize the cell nuclei, with confocal microscopy used to examine the samples (LSM700, Carl Zeiss, Jena, Germany).

### 4.9. Western Blot Analysis

Western blot analysis was done as previously described [2,6,48]. Proteins from cells were isolated, concentrations were determined, and samples were heated in sodium dodecyl sulfate for 5 min. Proteins samples were separated using sodium dodecyl sulfate-polyacrylamide gel electrophoresis. Proteins were deposited on poly (vinylidene fluoride) membranes and then blocked for 1 h with 5% skim milk at room temperature. Membranes were washed 3 times with TBST (washing buffer) for 10 min each time before being incubated overnight at 4 °C with primary antibodies (1:1000) against phosphorylated (p)-IκB (Cat. no: #2859), p-NF-κB, p-ERK (Cat. no: #9101S), total (T)-ERK (Cat. no: #9102S), p-JNK (Cat. no: #9251S), T-JNK (Cat. no: #9252), p-p38 (Cat. no: #9211), T-p38 (Cat. no: #9212), and β-actin ((Cat. no: #4967). The membranes were washed three times with TBST for 10 min each time and incubated with secondary antibody (1:3000) (Cat. no: #7074) for 1 h before being rinsed three times with TBST for 10 min each time. Protein bands were identified using enhanced chemiluminescence solutions (1:1 ratio) in an Imager ALS 500.

### 4.10. Statistical Analysis

Data are presented as the means ± standard error of the mean (SEM) (*n* = 3). One-way analysis of variance was used to establish statistical significance. The data was analyzed with GraphPad Prism 8.4.3 (GraphPad Software, LLC., San Diego, CA, USA). Probability values were considered significant at * *p* < 0.05, ** *p* < 0.01, and *** *p* < 0.001 in comparison to the CFA group, and at ^#^
*p* < 0.001 in comparison to the control.

## 5. Conclusions

Our study showed that isoleucilactucin has anti-inflammatory activities in CFA-induced MH-S cells. These findings add to our understanding of CFA-induced inflammatory responses and reveal that isoleucilactucin may decrease proinflammatory mediators and cytokines expression in murine alveolar macrophages, which is known as MH-S cells. Our results indicate that isoleucilactucin possesses an anti-inflammatory potential for inflammation management across a variety of inflammatory illnesses. Based on our results, the effects of isoleucilactucin on NF-κB and mitogen-activated protein kinase (MAPK) signaling have been linked to inflammation. More research is required to identify the exact molecular pathways in animal models that cause inflammation and determine how isoleucilactucin could be used as an herbal treatment to prevent inflammation.

## Figures and Tables

**Figure 1 ijms-22-09506-f001:**
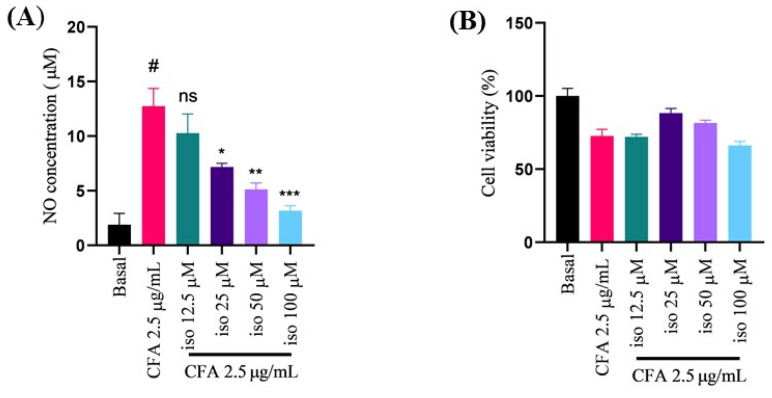
Effect of isoleucilactucin on coal fly ash (CFA)-induced nitric oxide (NO) generation and viability in MH-S cells. (**A**) Cells were categorized into six groups: basal (control), CFA (2.5 μg/mL), and four concentrations of CFA with isoleucilactucin (12.5, 25, 50, and 100 μM). Cells were treated with the different isoleucilactucin concentrations for 30 min before CFA treatment and incubated for 18 h. The Griess reagent method was used to determine the NO level. (**B**) The MTT reagent method was used to perform a cell viability experiment. A 24-well plate was used to seed the cells. Values from three independent experiments were expressed as the mean ± SEM (*n* = 3). Compared with the basal group, ns: not significant, ^#^
*p* < 0.001; compared with the CFA group, * *p* < 0.05, ** *p* < 0.01, and *** *p* < 0.001.

**Figure 2 ijms-22-09506-f002:**
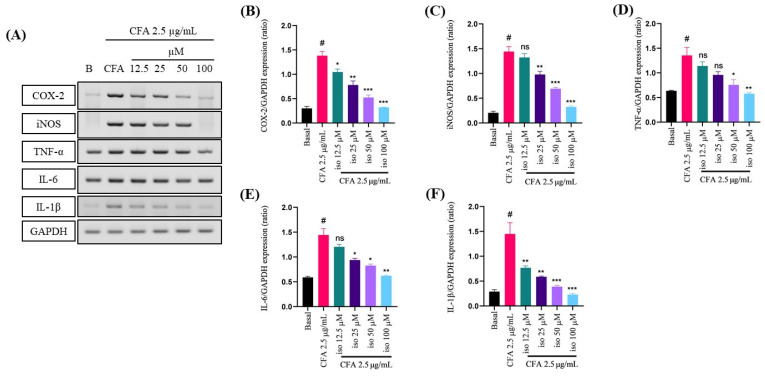
Effect of isoleucilactucin on CFA-induced proinflammatory mediators and cytokines in MH-S cells analyzed using reverse transcription polymerase chain reaction (RT-PCR). (**A**) RT-PCR analysis was performed to evaluate the mRNA levels of proinflammatory mediators, such as cyclooxygenase 2 (COX2) and inducible nitric oxide synthase (iNOS), as well as proinflammatory cytokines, such as tumor necrosis factor-alpha (TNF-α), interleukin (IL)-6, IL-1β, and GAPDH. (**B**–**F**) ImageJ was used to perform quantitative analysis of the relative mRNA expression levels. Isoleucilactucin dosages of 12.5, 25, 50, and 100 μM were used to seed the cells in a 6-well plate. Values from three independent experiments were expressed as the mean ± SEM (*n* = 3). Compared with the basal group, ns: not significant, ^#^
*p* < 0.001; compared with the CFA group, * *p* < 0.05, ** *p* < 0.01, and *** *p* < 0.001.

**Figure 3 ijms-22-09506-f003:**
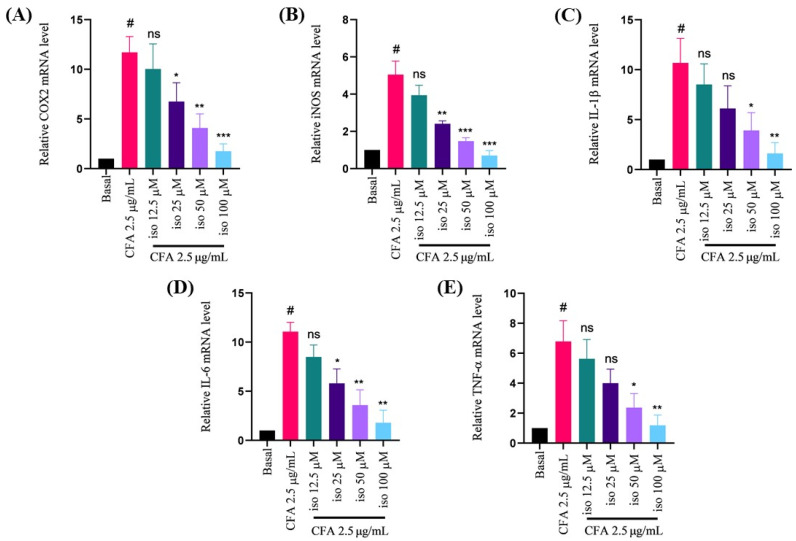
Effect of isoleucilactucin on CFA-induced mRNA expression in MH-S cells measured using real-time polymerase chain reaction (PCR). (**A**–**E**) The mRNA levels of COX2, iNOS, IL-1β, IL-6, and TNF-α were analyzed by PCR after 18 h of incubation with CFA. Isoleucilactucin dosages of 12.5, 25, 50, and 100 μM were used. GAPDH was used as a housekeeping gene. Values from the three independent experiments were expressed as the mean ± SEM (*n* = 3). Compared with the basal group, ^#^
*p* < 0.001; compared with the CFA group, ns: not significant, * *p* < 0.05, ** *p* < 0.01, and *** *p* < 0.001.

**Figure 4 ijms-22-09506-f004:**
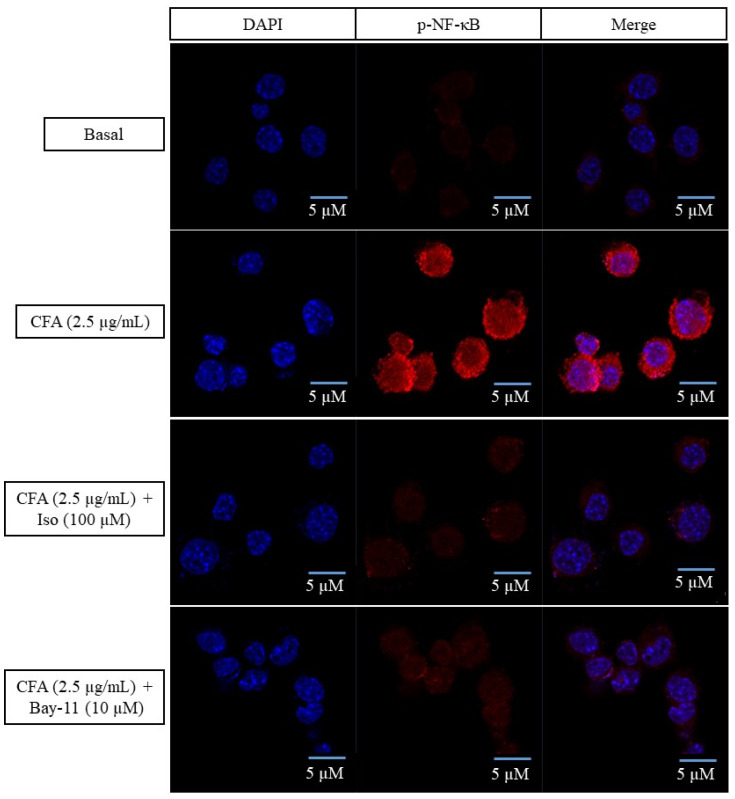
Effect of isoleucilactucin on CFA-induced nuclear factor-kappa B (NF-κB) translocation in MH-S cells examined by immunofluorescence (IF) assay. In a 6-well plate, cells were seeded on a coated cover slip and divided into four groups: basal, CFA, CFA with isoleucilactucin (100 μM), and CFA with Bay-11 (p-NF-κB inhibitor). Before treatment with CFA (2.5 µg/mL), cells were treated for 30 min with isoleucilactucin and Bay-11 (10 µM) and incubated for 18 h. IF staining was used to examine the p-NF-κB nuclear translocation. To observe the nuclei, the samples were mounted using a ProLong Gold Antifade Reagent containing DAPI (blue). Confocal microscopy at 1000× magnification was used to examine the stained cells.

**Figure 5 ijms-22-09506-f005:**
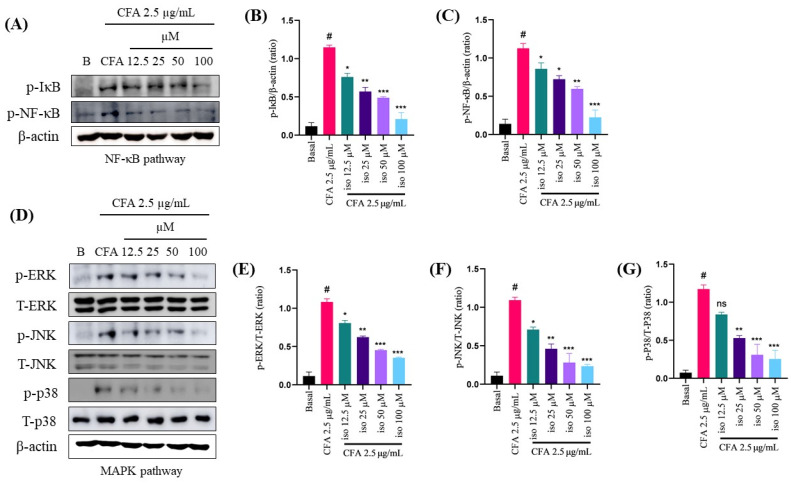
Effect of isoleucilactucin on CFA-induced NF-κB and mitogen-activated protein kinase (MAPK) pathways in MH-S cells measured by Western blotting. (**A**) Western blot analysis was done to evaluate the protein levels of the phosphorylated (p)-IκB and p-NF-κB pathways. (**B**,**C**) ImageJ was used to perform densitometric analysis of the protein expression levels. (**D**) Western blotting was used to investigate the MAPK pathways, which includes ERK, JNK, and p38. (**E**–**G**) ImageJ was used to perform densitometric analysis of the protein expression levels. As a loading control, β-actin was used. Isoleucilactucin dosages of 12.5, 25, 50, and 100 μM were used on cells seeded in a 6-well plate. Values from the three independent experiments were expressed as the mean ± SEM (*n* = 3). Compared with the basal group, ^#^
*p* < 0.001; compared with the CFA group, ns: not significant, * *p* < 0.05, ** *p* < 0.01, and *** *p* < 0.001.

**Figure 6 ijms-22-09506-f006:**
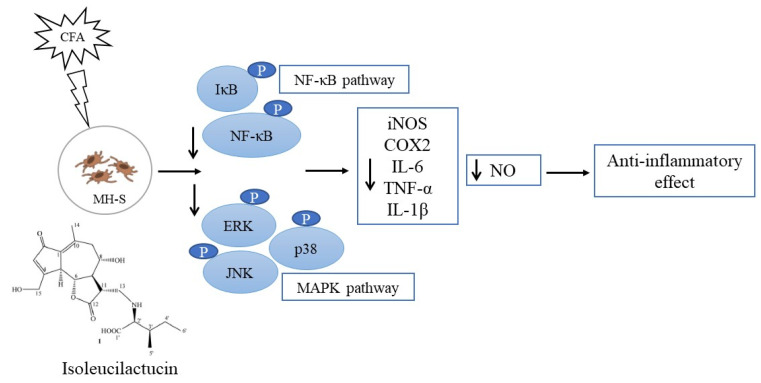
Proposed mechanistic model of the role of isoleucilactucin on CFA-induced inflammation in MH-S cells.

**Table 1 ijms-22-09506-t001:** Primers used for reverse transcription polymerase chain reaction (RT-PCR) and real-time PCR analysis in this study.

H	Primer Sequence
*GAPDH* *	F: 5′-CACTCACGGCAAATTCAACGGCAC-3′ R: 5′-GACTCCACGACATACTCAGCAC-3′
*iNOS* *	F: 5′-CCCTTCCGAAGTTTCTGGCAGCAGC-3′ R: 5′-GGCTGTCAGAGCCTCGTGGCTTTGG-3′
*COX-2* *	F: 5′-CACTACATCCTGACCCACTT-3′ R: 5′-ATGCTCCTGCTTGAGTATGT-3′
*TNF-α **	F: 5′-TTGACCTCAGCGCTGAGTTG-3′ R: 5′-CCTGTAGCCCACGTCGTAGC-3′
*IL-1β* *	F: 5′-CTGTGGAGAAGCTGTGGCAG-3′ R: 5′-GGGATCCACACTCTCCAGCT-3′
*IL-6* *	F: 5′-GTACTCCAGAAGACCAGAGG-3′ R: 5’-TGCTGGTGACAACCACGGCC-3′
*GAPDH* **	F: 5′-CACTCACGGCAAATTCAACGGCAC-3′ R: 5′-GACTCCACGACATACTCAGCAC-3′
*iNOS* **	F: 5′-GGCAGCCTGTGAGACCTTTG-3′ R: 5′-GCATTGGAAGTGAAGCGTTTC-3′
*COX-2* **	F: 5′-GGGTGTCCCTTCACTTCTTTCA-3′ R: 5′-TGGGAGGCACTTGCATTGA-3′
*TNF-α* **	F: 5′-TGCCTATGTCTCAGCCTCTTC-3′ R: 5′-GAGGCCATTTGGGAACTTCT-3′
*IL-1β* **	F: 5′-CAACCAACAAGTGATATTCTCCATG-3′ R: 5′-GATCCACACTCTCCAGCTGCA-3′
*IL-6* **	F: 5′-TCCAGTTGCCTTCTTGGGAC-3′ R: 5′-GTGTAATTAAGCCTCCGACTTG-3′

* Primer sequence for RT-PCR; ** Primer sequence for Real-Time PCR.

## Data Availability

Not applicable.

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
