# Peer review of "Isoleucilactucin Ameliorates Coal Fly Ash-Induced Inflammation through the NF-κB and MAPK Pathways in MH-S Cells"

_ijms, 2021, doi:10.3390/ijms22179506_

Round 1

Reviewer 1 Report

1.The detail of isoleucilactucin needs to be provided. It seems to be a newly found molecule. 
2. Animal study is strongly recommended.
3. MH-S cells cannot represent alveolar macrophages (AMs) based on a previous study (PMID: 26916143). Authors should confirm the current results in primary isolated AMs and human AMs. 
4. Authors should use other TLR ligands as a control. 
5. CFA in the title should be fully spelled out.  

Author Response

Respond Letter

Answer of the reviewer-1

1.The detail of isoleucilactucin needs to be provided. It seems to be a newly found molecule.

Answer: Isoleucilactucin, one of the major active constituents of Ixeridium dentatum, has been reported to have amylase secretion activity after treatment with high glucose in human salivary gland cells.

The leaves of Ixeridium dentatum collected in the area of Jeonju-si, Jeonbuk, Korea in May 2017 and were certified by Professor Sang-Won Lee of the National Institute of Horticultural and Herbal Science, South Korea. A voucher specimen is kept in the Herbarium of the College of Pharmacy, Yonsei University, Incheon, Korea (ID201705).

The plant extraction and isolation were conducted as previously described [20,47]. In brief, dried leaves of I. dentatum (5.0 kg) were extracted with MeOH (3 × 10 L, 50°C) and sonicated for 4 h after solvent evaporation, producing 290.0 g extract. Extract was suspended in H2O and partitioned with n-hexane and EtOAc to obtain n-hexane (ID1, 91.0 g), EtOAc (ID2, 15.1 g), and H2O (ID3, 178.0 g) extracts after the solvents were removed in vacuo. In a Diaion HP-20 column, the H2O fraction was chromatographed with increasing MeOH concentrations (25, 50, and 75 %) to acquire three sub-fractions: ID3A (14.5 g), ID3B (13.3 g), and ID3C (23.2 g). To obtain isoleucilactucin, the ID3C fraction was chromatographed on an RP-18 CC and eluted with MeOH:H2O (1:2, v/v) (5.2 mg). The resulting isoleucilactucin was a white amorphous powder. The HR-ESI-MS [M–H2O]+ ion at m/z 389.1825 identified its molecular formula as C21H29NO7 (calcd for C21H27NO6, 389.1838).

  1. Animal study is strongly recommended.

Answer: In this manuscript, we did the experiment in in vitro due to lower amount of single compound and also later, we have plan to do large scale experiment including animal study.

  1. MH-S cells cannot represent alveolar macrophages (AMs) based on a previous study (PMID: 26916143). Authors should confirm the current results in primary isolated AMs and human AMs.

Answer: Based on literature and our study MH-S cells represent lung macrophages. Here, we provided some supporting information. This is our first in vitro investigation, and we intend to expand to primary isolated AMs and human AMs in the future project.

https://www.nature.com/articles/s41598-020-68965-5

https://www.hindawi.com/journals/ecam/2021/5546052/

https://www.thermofisher.com/kr/ko/home/technical-resources/cell-lines/m/cell-lines-detail-620.html

https://www.atcc.org/products/crl-2019#:~:text=The%20MH%2DS%20cell%20line,population%20of%20mouse%20alveolar%20macrophages.&text=The%20cells%20retain%20many%20of,esterase%20positive%20and%20peroxidase%20negative.

  1. Authors should use other TLR ligands as a control.

Answer: Here, we showed the NO induction by LPS to compare with CFA. We got the very similar result in CFA and LPS treated group.

  1. CFA in the title should be fully spelled out.

Answer: We have written the full name of CFA as coal fly ash.

Reviewer 2 Report

Major concerns

  1. Unless the compound under evaluation (isoleucilactucin) is intended to be used exclusively in animals, it is not justified to use only a murine cell line knowing that there are dozens of human macrophage cell lines available on the market. Alternatively, fresh monocyte-derived macrophages (MDMs) could be recovered from healthy donors.
  2. Throughout the manuscript, nothing is mentioned about the solubility of isoleucilactucin. Is it sufficiently water-soluble to be available in the culture medium to exert its anti-inflammatory activity at cellular level? Is anything known about its stability in solution? Have studies been done on animal-derived fluids?
  3. Beyond the mechanisms described in this manuscript, the authors should indicate, suggest or hypothesize how isoleucilactucin acts at the membrane level, any receptor involved? Does it permeate the cell?
  4. What is the rationale for using the CFA concentration used (2.5 ug/ml) throughout the manuscript? Similarly, and more importantly, the four concentrations of isoleucilactucin used in the manuscript, where did they come from? Is it known if they are physiological concentrations well tolerated by an animal model?
  5. This reviewer considers that this manuscript would significantly benefit from including a mechanistic model figure that recapitulates the mechanisms presented in the study. It is strongly suggested that it should be included, it would certainly help readers to better understand the main findings of the paper.
  6. The catalog numbers of the antibodies used must be included (lines 267, 278, and 281).

Minor concerns

  1. Verify typos on lines 15, 65, and 193.
  2. After the following sentence: "Effects of isoleucilactucin on NF-κB and mitogen-activated protein kinase (MAPK) signaling have been linked to inflammation" a citation must be included.
  3. As indicated in Figure 4, the confocal studies were done only at one concentration (100 uM) of isoleucilactucin. Did the other concentrations generate similar results? This point should be discussed.
  4. The scale bars are not visible in Figure 4, they must be fixed.
  5. On line 206, is it RPMI or RPMI-1640? Please, check.
  6. The models of the instruments mentioned in lines 226, 239, 246, and 270 must be indicated.
  7. The DMSO concentration on line 245 and the ethidium bromide concentration on line 255 should be mentioned.
  8. The source of the primers is not indicated, it must be indicated.
  9. Indicate the temperature that was used for blocking (line 264) and incubation of the IgG Fab2 (line 267).
  10. In the Statistical Analysis section, the software used to analyze the data must be indicated.

Author Response

Respond Letter

Answer of the reviewer-2

Major concerns

  1. Unless the compound under evaluation (isoleucilactucin) is intended to be used exclusively in animals, it is not justified to use only a murine cell line knowing that there are dozens of human macrophage cell lines available on the market. Alternatively, fresh monocyte-derived macrophages (MDMs) could be recovered from healthy donors.

Answer: Thank you very much for your wonderful feedback. We looked at the anti-inflammatory properties of isoleucilactucin in MH-S cells as a first step. According to our findings, isoleucilactucin has anti-inflammatory characteristics. We intend to conduct experiments in animals and additional macrophage cell lines in our future research to demonstrate the exact mechanism of isoleucilactucin in inflammation.

  1. Throughout the manuscript, nothing is mentioned about the solubility of isoleucilactucin. Is it sufficiently water-soluble to be available in the culture medium to exert its anti-inflammatory activity at cellular level? Is anything known about its stability in solution? Have studies been done on animal-derived fluids?

Answer: In our study, isoleucilactucin is dissolved in DMSO. It completely dissolves in DMSO. There is not enough data on solution stability. There have been no investigations of isoleucilactucin on animal-derived fluids to our knowledge. We will test it out in our future project.

  1. Beyond the mechanisms described in this manuscript, the authors should indicate, suggest or hypothesize how isoleucilactucin acts at the membrane level, any receptor involved? Does it permeate the cell?

Answer: Based on our results, we hypothesize that isoleucilactucin permeate the cell. Because, treatment with CFA increased NF-κB translocation from the cytoplasm to the nucleus, but treatment with the maximum dose of isoleucilactucin (100 μM) inhibited p-NF-κB nuclear translocation. Additionally, pretreatment with isoleucilactucin greatly decreased the p-IκB and NF-κB in MH-S cells, whereas treatment with CFA significantly enhanced the p-IκB and p-NF-κB in alveolar macrophages. Furthermore, as compared to treatment with CFA alone, MAPK pathways, including p-ERK, p-JNK, and p-P38, were dose-dependently reduced. However, more research is required to establish the precise mechanism of isoleucilactucin.

  1. What is the rationale for using the CFA concentration used (2.5 ug/ml) throughout the manuscript? Similarly, and more importantly, the four concentrations of isoleucilactucin used in the manuscript, where did they come from? Is it known if they are physiological concentrations well tolerated by an animal model?

Answer: To begin, we obtained the highest nitric oxide (NO) induction without cytotoxicity at CFA (2.5 ug/ml) in our primary screening study. Lower doses resulted in minimal NO induction, but greater doses resulted in cytotoxicity. Second, based on our primary screening results, we employed isoleucilactucin (12.5, 25, 50, and 100 μM). Finally, there is no investigation of physiological isoleucilactucin concentrations on animal models in the literature.

  1. This reviewer considers that this manuscript would significantly benefit from including a mechanistic model figure that recapitulates the mechanisms presented in the study. It is strongly suggested that it should be included, it would certainly help readers to better understand the main findings of the paper.

Answer: Yes, we have added the model mechanism in the manuscript.

  1. The catalog numbers of the antibodies used must be included (lines 267, 278, and 281).

Answer: We provided the catalog numbers of the all antibodies in the manuscript.

Minor concerns

  1. Verify typos on lines 15, 65, and 193.

Answer: We have checked and corrected in the manuscript.

  1. After the following sentence: "Effects of isoleucilactucin on NF-κB and mitogen-activated protein kinase (MAPK) signaling have been linked to inflammation" a citation must be included.

Answer: This statement is based on our findings. As a result, this statement is now included in the conclusion.

  1. As indicated in Figure 4, the confocal studies were done only at one concentration (100 uM) of isoleucilactucin. Did the other concentrations generate similar results? This point should be discussed.

Answer: In confocal study, we examined the highest concentration of isoleucilactucin (100 uM). Because, this concentration showed the maximum inhibition compared to the Bay-11. That’s why, we checked for only maximum dose of isoleucilactucin.

  1. The scale bars are not visible in Figure 4, they must be fixed.

Answer: We added visible scale bars in the figures 4.

  1. On line 206, is it RPMI or RPMI-1640? Please, check.

Answer: This is RPMI-1640.

  1. The models of the instruments mentioned in lines 226, 239, 246, and 270 must be indicated.

Answer: We have provided the models of the instruments in the manuscript.

  1. The DMSO concentration on line 245 and the ethidium bromide concentration on line 255 should be mentioned.

Answer: The concentrations of DMSO and ethidium bromide were 100% and 0.006% respectively.

  1. The source of the primers is not indicated, it must be indicated.

Answer: The primers were obtained from Bioneer, Daejeon, Republic of Korea.

  1. Indicate the temperature that was used for blocking (line 264) and incubation of the IgG Fab2 (line 267).

Answer: Using 2% BSA, cells were blocked for 1 h at room temperature before being incubated overnight at 4°C with the primary antibody rabbit anti-p-NF-κB. Samples were then incubated with secondary antibody, IgG Fab2 for 1 h in the dark at room temperature.

  1. In the Statistical Analysis section, the software used to analyze the data must be indicated.

Answer: We have indicated the software name in the statistical analysis section. The data was analyzed with GraphPad prism 8.4.3.

Round 2

Reviewer 1 Report

There is no more comment. However, additional experiments using an animal model and human cells are needed.